# Development and Validation of a Novel Ultra-Compact and Cost-Effective Device for Basic Hands-On CPR Training: A Randomized, Sham-Controlled, Blinded Trial

**DOI:** 10.3390/ijerph192215228

**Published:** 2022-11-18

**Authors:** Joseba Rabanales-Sotos, Isabel María Guisado-Requena, Zoila Esperanza Leiton-Espinoza, Carmen María Guerrero-Agenjo, Jesús López-Torres-Hidalgo, José Luis Martín-Conty, Francisco Martín-Rodriguez, Jaime López-Tendero, Angel López-González

**Affiliations:** 1Department of Nursing, Physiotherapy and Occupational Therapy, Facultad de Enfermería de Albacete, University of Castilla-La Mancha (Universidad de Castilla-La Mancha/UCLM), Campus Universitario s/n, 02071 Albacete, Spain; 2Group of Preventive Activities in the University Health Sciences Setting, University of Castilla-La Mancha (Universidad de Castilla-La Mancha/UCLM), Campus Universitario s/n, 02071 Albacete, Spain; 3Nursing Faculty, National University of Trujillo (Universidad Nacional de Trujillo), Trujillo 130101, Peru; 4Castilla-La Mancha Health Service (Servicio de Salud de Castilla-La Mancha/SESCAM), University of Castilla-La Mancha (Universidad de Castilla-La Mancha/UCLM), 02071 Albacete, Spain; 5Albacete Faculty of Medicine, Castilla-La Mancha Health Service (Servicio de Salud de Castilla-La Mancha/SESCAM), University of Castilla-La Mancha (Universidad de Castilla-La Mancha/UCLM), 02071 Albacete, Spain; 6Faculty of Health Sciences, University of Castilla-La Mancha (Universidad de Castilla-La Mancha/UCLM), 13001 Ciudad Real, Spain; 7Advanced Clinical Simulatons Center, School of Medicine, Universidad de Valladolid, 47002 Valladolid, Spain

**Keywords:** basic cardiac life support, heart massage, first aid, health knowledge, attitudes, practice

## Abstract

To examine the performance of a novel low-cost, ultra-compact, and attractive auditory feedback device for training laypeople in external chest compressions (ECCs), we conducted a quasi-experimental cross-sectional study from September to November 2021 at the Faculty of Nursing of Albacete, University of Castille-La Mancha, Spain. The ECC sequence was performed in the laboratory with the new device for basic hands-on CPR training. Results: One hundred college students were included in this study. The compression rate/min with the new device was 97.6, and the adequate %ECC was 52.4. According to the status of body mass index (BMI) and muscle strength of the upper limbs in the bivariate analysis, it was observed that the new device discriminated between those who performed correct ECCs according to their BMI and muscle strength and those who did not, which led to significantly influenced results in terms of the percentage of ECCs with correct depth. Conclusions: The new ultra-compact auditory feedback device “Salvando a Llanetes^®^” demonstrated utility for teaching and learning ECCs in basic CPR. We can affirm that the analyzed device is an adequate, safe and economical method for teaching “CPR Hands-Only™” to the general population.

## 1. Introduction

The annual incidence of out-of-hospital cardiopulmonary arrest (OHCA) is 55/100,000 inhabitants/year, with 70% originating at home and 20% in public settings. Roughly 20% of OHCAs are resuscitated by laypersons [1]. The immediate initiation of cardiopulmonary resuscitation (CPR) by a citizen improves survival for OHCA, and it is estimated that early CPR performed by a witness can double or triple the chances of survival. Quick identification and early cardiopulmonary resuscitation (CPR) is critical to maximize survival [2]. Basic CPR teaching must encourage as many people as possible to join the chain of survival at the soonest possible time. Therefore, designing strategies for teaching and learning will increase the chances of survival of OHCA [2,3].

Organizations such as the International Liaison Committee (ILCOR), the American Heart Association (AHA) and the European Resuscitation Council (ERC) advocate for simplification and universality of basic CPR; for this purpose, they have implemented plans like “CPR Hands-Only™” [4,5]. In addition, the ongoing COVID-19 pandemic has changed resuscitation workflows, mainly influenced by risks of contamination during mouth-to-mouth ventilation [6,7]. Consequently, current guidelines recommend the rapid application of chest compressions up until the arrival of emergency medical services (EMS) personnel [8].

New methodologies and teaching materials have been proposed for teaching CPR to improve training and retention of what has been learned. In addition, this methodology indicates that evaluation instruments must be available to assess knowledge and skills. The teaching methodology and materials used must consider the characteristics of each student in their cognitive, motor and socio-affective maturity and use active methodologies such as learning-by-doing, with demonstration and redemonstration [9,10]. The training should include practical training complemented by theoretical aspects, including virtual teaching. This training can be carried out without sophisticated material skills [10]. However, the best training method for each population group remains to be defined [11,12]. For this reason, new training strategies with more active methodologies are being explored, including self-learning through low-cost feedback devices. This feedback increases the quality of CPR both in training and during CPR performance [13]. Evidence supports the use of real-time feedback to develop chest compression skills [14].

The issue is knowing how to train the largest potential pool of responders in the shortest possible timescale and with the optimal use of resources [15]. Efforts to disseminate knowledge of basic CPR are ongoing and encompass several areas, e.g., analysis of mastery learning vs. time-based education, self-learning basic life support, the use of virtual reality mobile applications, and appropriate training started at a young age [16,17,18,19]. All the previously mentioned methodological aids are useful ways to facilitate widespread diffusion of basic CPR; however, effective CPR training requires a hands-on approach. In this sense, low-cost, portable, rugged, and tested training devices are essential to provide training on a mass scale or in wilderness or under-resourced locations.

The scope of this study aimed to examine the performance of a novel low-cost auditory feedback device for training laypeople in external chest compressions (ECCs).

## 2. Materials and Methods

### 2.1. Materials

-Salvando a Llanetes^®^ (Cardioprotec&Health, Spain): This is an ultra-compact and extremely economical simulator for basic hands-on CPR training. This is a simple device composed of an auditory feedback heart in plastic material (60 × 60 × 50 mm) that, when compressed perpendicularly and with sufficient force, emits a sound due to the mobilization of the air contained in an interior cavity. This device does not require batteries (Figure 1). The heart is placed on an anthropomorphic figure drawn on a tapestry, on which the basic ERC-CPR algorithm is printed (Figure 2) [5]. The heart, made of plastic material by the Winther-Winther company—Copenhagen, Denmark—is suitable for teaching ECC. It has a hardness of 7 on the Shore D scale s [20] and an auditory feedback system that emits a sound of 65–75 dB in intensity and 6000–8000 Hz in frequency when pressed correctly. The characteristics of hardness and the pressure with which the heart must be compressed are necessary to reach the 50–60 mm described as the adequate depth for ECC in adults [21].-Bruel & Kjaer Integrating Impulse Sound Level Meter Type 2226 (Brüel & Kjær Sound & Vibration Measurment A/S, Nærum, Denmark).-Cell dynamometer of load SENSYs 5962 (Sensy S.A. Charleroi, Belgium).

### 2.2. Study Design and Participants

For convenience, first-semester students studying nursing, between 18 and 25 years old, were included during the academic year 2021–2022, without prior knowledge or experience in CPR. Exclusion criteria included having previously received CPR training, suffering from cardiovascular and orthopedic disease, dysfunction that contraindicated or prevented performing CPR maneuvers, or refusal to participate in the study after knowing its objectives. No student met the exclusion criteria.

Participants were informed about the objectives and method of the study as well as their personal contributions to it. Following their acceptance of participation in the study, the participants (in groups of 10) received, 2–4 days before the measurement of the anthropometric variables and physical fitness, a session of standardized training in basic CPR based on the 2021 ERC guidelines [5]; COVID-19 recommendations were followed [22]. This training, supervised by an instructor who followed the CPR Personal Anytime Learning Programs method, allowed every student to practice with a mannequin, following a DVD presentation providing instructions as well as amendments from the instructor. After this, anthropometric and physical variables were measured.

All the participants were informed in detail about the nature of this study and provided written informed consent. The study protocol was approved, according to the Helsinki Declaration, by the Clinical Research Ethics Committee of the Albacete Health Area, Spain.

### 2.3. Method

A quasi-experimental pre–post study was conducted from September to November 2021 at the Faculty of Nursing of Albacete, part of the University of Castilla-La Mancha (UCLM), Spain. Between 2 and 4 days after the training session, the participants took the tests.

### 2.4. Variables

In addition to socio-demographic variables (age, sex, education, and residence), we measured the following in all the subjects:

-Weight. Average of two determinations measured by a certified Seca-770 scale of easy calibration, with the participant barefoot and in light clothes.-Height. Average of two determinations measured by wall mounted height rod Seca-222, with the participant barefoot in a standing position, and joining their sagittal average line with the height rod average line.-Body mass index. Calculated as weight (kg) by height (m^2^).-Muscle strength: The maximum strength of the upper body (capacity to produce the maximum muscular tension with a muscle contraction) evaluated with a digital hand grip dynamometer Takei TKK 5101 (rank 5–100 kg, accuracy 0.1 kg), which measures the force of maximum grip strength in both hands (with previous grip adjustment of the dynamometer based on the hand size), by alternatively making two attempts with each hand, with the subject standing up and resting their arm parallel to their body. The final score was the mean of the four measures (kg).-External chest compressions: Considered as adequate when (a) the rate was 100–120 compressions/min (measured by synchronization) [23]; (b) the heart was pressed with 28.5–69 kp of force (measured with a dynamometer placed over the heart) [18]; and (c) the heart emitted a sound of 65–75 dB intensity and 6000–8000 Hz frequency (measured with a sound level meter placed at a distance of 30 cm).-Adequate ECC (%): Adequate depth ECC (%) + adequate compression rate (%)/2 [24].-Opinion on the usefulness of the new device: Question with a dichotomous answer (yes/no).

### 2.5. Evaluation of Participants’ Cardiopulmonary Resuscitation Ability

Test: Participants performed two ECC minutes on the heart of the “Salvando a Llanetes^®^” auditory feedback device. The study flow diagram is shown in Figure 3.

Measurements were obtained at minute one and minute two. The test ended once the participants reached the objective (2 min of ECC).

During the tests, the participants received no explanation or corrections from the instructors. After the tests, participants were asked their opinion on the utility of the “Salvando a Llanetes^®^” device.

### 2.6. Statistical Analysis

All measurements were taken under standard conditions by the same researchers. Statistical normality of the variables was tested using both graphical (normal probability plot) and statistical procedures (Kolmogorov–Smirnov test). Differences by sex in demographic, anthropometric and muscle strength variables were tested using Student’s *t* test. All variables fit acceptably to a normal distribution. Using the chi-square test, the sex differences between the BMI categories were compared.

To assess whether the new device was capable of discriminating in favor of adequate ECC performance by the participants according to their BMI and muscle strength, BMI was categorized according to the age and gender cut-off points defined by the World Health Organization as underweight (<18.5 kg/m^2^), normal weight (18.5–24.9 kg/m^2^) and overweight/obesity (≥25 kg/m^2^). Muscular strength was dichotomized, taking the cut-off point of the percentile 25, by sex. Differences in the %ECC by categories of weight status (underweight vs. normal weight/overweight) and muscular strength (lower vs. higher) were tested using Analysis of Variance (ANOVA).

The variables were described using measures of central tendency (mean) and dispersion (standard deviation).

The criterion for bilateral statistical significance was set at *p* ≤ 0.05. All the statistical analyses were performed with the IBM^®^ SPSS^®^ Statistics 24 software.

## 3. Results

This study involved one hundred students, including 50% women, whose mean age was 19.8 years (SD: 3.1). Table 1 displays the anthropometric and muscular fitness characteristics by gender. The mean values of weight (73.8–16.0) vs. (59.0–11.6), height (1.74–0.05) vs. (1.63–0.06), BMI (24.2–5.1) vs. (22.3–3.98), and muscle strength of the upper limbs (40.2–6.9) vs. (24.3–4.1) were significantly higher in men than in women (*p* < 0.05). In weight status, significant differences were only observed in obesity, being greater in men than in women 16.7 vs. 4.0 (*p* < 0.05).

The quality ECC manoeuvres for sex (rhythm and adequate depth) at 2 min are shown in Table 2. Mean percentage of adequate compression depth was significantly higher in men than in women, 86.0 (71.7–94.3) vs. 43.7 (34.6–52.9) (*p* < 0.001), and mean percentage of adequate ECC was significantly higher in men than in women, 77.8 (65.1–90.5) vs. 41.2 (32.2–50.1) (*p* < 0.001).

Adequate %ECC was achieved according to weight status and muscle strength in bivariate analysis (Table 3). It was observed that in the “Salvando a Llanetes^®^” device, the BMI (34.20–29.04) vs. (70.92–25.29) and muscle strength of the upper limbs (28.82–40.93) vs. (59.44–29.09) significantly influenced the results.

In addition, we found that 88% of participants rated the feedback device positively, answering “Yes” to the question “Do you think the device used was useful?”.

## 4. Discussion

This study analyzes whether the proposed ultra-compact and economic device for basic hands-on CPR training is useful for performing ECCs in CPR and whether it can discriminate who will perform adequate ECCs according to BMI and muscle strength. Overall, the data show the percentage of ECC performance over 2 min reached adequate quality values, although lower than those previously reported with other high/medium fidelity simulators and at a much higher cost. These quality values, as with high/medium fidelity simulators, are also influenced by BMI and upper body muscle strength level.

Our results show that the alternative device “Salvando a Llanetes^®^” is useful for teaching “CPR Hands-Only™” to groups of laypeople by incorporating a feedback mechanism that allows testing of the depth of the compression in a very simple way, which coincides with other studies in which other, more sophisticated feedback devices were used [12,21,25,26,27,28,29,30]. In our study, the adequate %ECC achieved in the proposed ultra-compact and economic device did not reach the values achieved in high/medium fidelity simulators reported by others [12,21,25,26,27,28].

Taking into account that CPR must be taught to the general population from an early age [10,31], it is difficult to find a simple, inexpensive training device that is easy to store, and is not rejected by younger people. Although there are multiple simulation models on the market with different levels of complexity and measurement of results, the one used in this study is characterized as being simple, inexpensive and with results suitable for any age group, as well as having a pleasant nature that may be attractive to young people.

It seems clear that children under 12 old years of age have difficulty performing some of the practical skills involved in learning basic CPR techniques due to their physical characteristics. However, these younger children are capable of learning these and other theoretical issues related to life support (access to the emergency medical system, awareness, etc.), so it is possible to implement training by levels, starting with basic issues in the stage of early childhood, and gradually introducing knowledge and skills according to age and training cycle [32]. “Salvando a Llanetes^®^” is shown as an adequate device to introduce people to the steps that are part of emergency assistance (security in the area, request for help, request for an AED, etc.) and the practice of ECCs. The level of knowledge that can be transmitted by the alternative device is different depending on the age of the student, which makes it ideal for training at an early age.

The increase in CPR training activities offered by health science students to the general population explains in part the increase in rates in which CPR was performed by laypersons for OHCA. However, no studies on cost evaluation of the teaching materials used were found in the literature. The initial cost of acquiring these materials can be high and sometimes prohibitive [10,22,28,31,32,33,34]. In this sense, the technical simplicity of the “Salvando a Llanetes^®^” simulator can adjust costs to extremely low levels of accessibility.

The ability to perform adequate ECCs increases in line with BMI; a BMI ≥ 18.5 in adults has been identified as the success threshold, which our sample mostly matches. Some studies have indicated that upper body muscle strength levels in rescuers can ensure that adequate ECCs will be performed [20,35,36,37]. Our data confirm that, with this new device, average upper body muscle strength levels are independently beneficial in achieving a higher percentage of adequate ECCs in short resuscitations. Therefore, according to BMI status and muscle strength, it was observed that the heart of “Salvando a Llanetes^®^” discriminates between those who do not do compressions with adequate depth and those who do.

Teaching only ECCs in CPR to the population was justified, until January 2020, as an acceptable method to increase the number of OHCAs served by the population [4,5,10,19,27,31]. In the current socio-sanitary framework, with serious risk of abandonment of assistance to victims in cardiac arrest in domestic, work, and social environments, the “Salvando a Llanetes^®^” device, by achieving a high percentage of correct ECCs, is suitable for citizens to learn CPR in large training groups [5,6,22]. As a device that bases training on the “CPR Hands-Only™” method, it is safe against the transmission of diseases, including COVID-19.

## 5. Strengths and Limitations

“Salvando a Llanetes^®^” is an extremely simple, disposable device with limitations on use in complex situations and spaces (water, rain, the need for high-fidelity simulations, etc.), which could mean a limitation in learning quality CPR. However, the same characteristics, together with its low economic cost, lightness, ease of transport and appealing appearance, make it appropriate for large population groups and/or low socio-economic status groups to start learning CPR from very early ages.

Another limitation could be that the results obtained were due to the type of sample, since nursing students, even if they are laymen, have a particular predisposition towards learning health-related procedures. Future studies should consider other populations such as schoolchildren, older adults, etc.

## 6. Conclusions

Training in CPR must begin at an early age, with age-appropriate methodologies and materials and accessibility to training spaces, with periodic retraining that includes recognition of cardiac arrest, request for help to EMS and the introduction of more complex actions with increasing age and size. However, the best training system remains unclear.

The low economic cost and small dimensions of this new device make it very accessible to population groups that do not have access to other higher quality simulators due to manufacturing costs and limitations of use in areas of limited accessibility.

We can affirm that the analyzed device “Salvando a Llanetes^®^” is a safe device for teaching “CPR Hands-Only™” to the general population.

In light of the results obtained and after verifying the adequacy of the new auditory feedback devices to teach basic CPR, further research is suggested, which can show results in terms of usefulness of the device in teaching CPR in places of limited accessibility and low levels of economic development.

Future research should prove the effectiveness in prolonged CPR, in which it would be necessary to rotate rescuers (5, 10 min and more).

## Figures and Tables

**Figure 1 ijerph-19-15228-f001:**
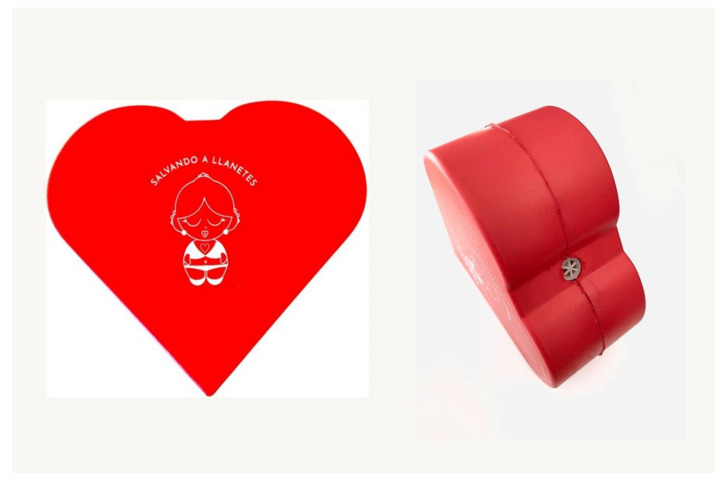
Heart of the auditory feedback device “Salvando a Llanetes^®^”.

**Figure 2 ijerph-19-15228-f002:**
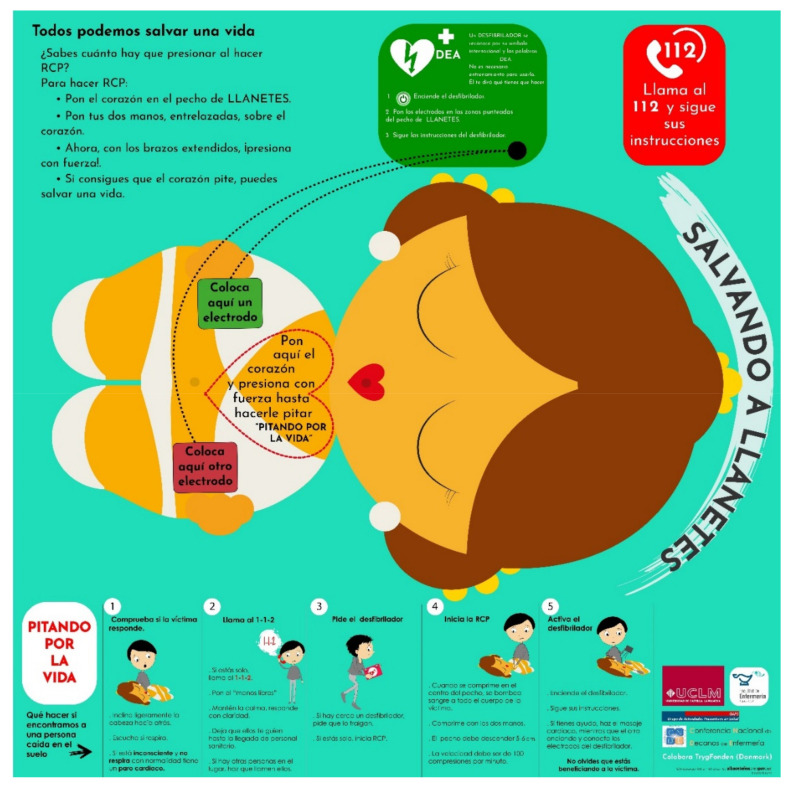
Auditory feedback device base for “Salvando a Llanetes^®^”.

**Figure 3 ijerph-19-15228-f003:**
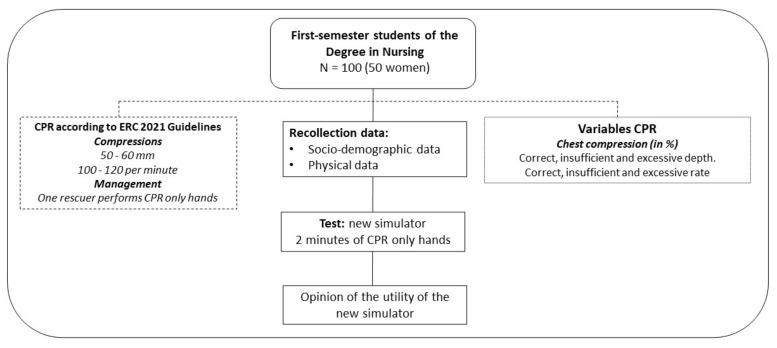
Study flow diagram.

**Table 1 ijerph-19-15228-t001:** Demographic, anthropometric and physical fitness variables of study population, by sex.

	Total (*n* = 100)	Men (*n* = 50)	Women (*n* = 50)	*p*
Age (years)	19.8 (3.1)	21.0 (4.6)	19.4 (2.5)	**0.022**
Weight (kg)	61.9 (13.8)	73.8 (16.0)	59.0 (11.6)	**<0.001**
Height (m)	1.67 (0.08)	1.74 (0.05)	1.63 (0.06)	**<0.001**
Body mass index (kg/m^2^)	22.7 (4.2)	24.2 (5.1)	22.3 (3.9)	**<0.045**
Weight status (%)				
Underweight	9.6	4.2	10.9	0.315
Normal weight	72.0	70.8	72.3	0.887
Overweight	12.0	8.3	12.9	0.539
Obesity	6.4	16.7	4.0	**0.022**
Muscular fitness (Handgrip dynamometry, kg)	27.4 (7.9)	40.2 (6.9)	24.3 (4.1)	**<0.001**

Values are means ± standard deviation, except for prevalence weigh status (%). In bold type: *p* ≤ 0.05.

**Table 2 ijerph-19-15228-t002:** New device quality of ECC by sex.

	Total (*n* = 100)	Men (*n* = 50)	Women (*n* = 50)	*p*
**Compression Rate/min**	97.6 (96.3–98.9)	99.1 (97.7–100.5)	97.1 (95.4–98.8)	0.198
**Adequate Depth ECC (%)**	56.7 (48.4–65.1)	86.0 (71.7–94.3)	43.7 (34.6–52.9)	**<0.001**
**Adequate ECC (%)**	52.4 (44.1–60.7)	77.8 (65.1–90.5)	41.2 (32.2–50.1)	**<0.001**

Values are mean ± confidence interval 95%. In bold type: *p* ≤ 0.05.

**Table 3 ijerph-19-15228-t003:** New device showed adequate ECC by categories of body mass index and muscular fitness.

**BMI**		
Underweight (*n* = 8)	34.20 (29.04)	
Normal Weight/Overweight (*n* = 92)	70.92 (25.29)	**<0.001**
**Muscular strength**		
Lower (*n* = 24)	28.82 (40.93)	
Higher (*n* = 76)	59.44 (29.09)	**0.011**

Categories of muscular fitness are Lower = 1st quartile and Higher = 2nd, 3rd and 4th quartiles by age and sex. Values are means ± standard deviation. In bold type: *p* ≤ 0.05.

## Data Availability

The data can be requested from the correspondence author.

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
