# Peer review of "Development and Validation of a Novel Ultra-Compact and Cost-Effective Device for Basic Hands-On CPR Training: A Randomized, Sham-Controlled, Blinded Trial"

_ijerph, 2022, doi:10.3390/ijerph192215228_

Round 1

Reviewer 1 Report

Congratulations to the authors for this interesting article. The future of CPR lies in mass dissemination, combating the high-cost barrier of traditional manikins.

The paper is generally well-written and presents novel conclusions. This article will be referenced in the scientific literature related to the teaching of CPR.

Only two considerations. VALIDATION is mentioned in the title, but I don't think this is the right term. This material is not validated, it is TESTED/ FEASIBILITY is analyzed. My title suggestion would be: “Development and feasibility of a novel ultra-compact and cost-effective device for basic hands-on resuscitation training. A randomized, sham-controlled, blinded trial”

Better "resuscitation" not "CPR" in the title.

On the other hand, in the limitations, I would point out that the good results obtained could be considered by the type of sample, since nursing students, even if they are laymen, have a particular predisposition towards learning procedures related to health. Future studies should contemplate other populations such as schoolchildren, elderly adults, etc.

Author Response

Thank the reviewer´s comments and suggestions. All have been considered and if appropriate incorporated into the manuscript. We believe that through all the comments and suggestions have greatly improved our manuscript.

We have sent a detailed, point-by-point response to the reviewers' comments.

Reviewer 2 Report

the authors present a novel low cost and portable device in training adequate CPR in younger people 

the study cohort was effectively assessed in their use of this novel CPR training tool 

this study is well designed and interesting and merits publication 

the general scientific interest in this topic will be rather limited and the applicability and scalability of the findings will be limited as well as the study cohort was small in quantity 

- I recommend the authors comment on the assessment of prolonged CPR and rotating CPR and effectiveness with participant fatigue at 5 min and 10 min -- this may be a topic for future study 

- I recommend the authors include a table with currently available similar CPR training tools with size and weight and portability and cost as a comparison to their novel device proposed 

Author Response

(The authors gave the same response as above.)
